

# Retrospective study on the correlation between serum MIF level and the condition and prognosis of patients with traumatic head injury

Zhentong Liu, Chengwu Liu and Kegao Ma

The Emergency Department, Qingdao Chengyang District People's Hospital, Qingdao, China

## ABSTRACT

**Objective:** This study aimed to investigate the correlation between serum levels of macrophage migration inhibitory factor (MIF) and the condition and prognosis of patients with traumatic brain injury (TBI).

**Methods:** A retrospective study design was used, and the clinical data of 131 TBI patients from February 2019 to January 2022 were analyzed. Patients were divided into mild (13–15 points), moderate (9–12 points), or severe (3–8 points) groups according to their Glasgow Coma Scale (GCS) score after admission. The serum levels of BDNF, MIF, and MBP in the three groups were compared, and their correlation with the severity of TBI was analyzed. Patients were then separated into a good prognosis group (4–5 points) and a poor prognosis group (≤3 points) based on their Glasgow Prognostic Score (GOS) after 6 months of follow-up. The predictive power of serum indexes and combined detection on prognosis was analyzed.

**Results:** Patients were classified into a mild group ($n$ = 63), moderate group ($n$ = 47), and severe group ($n$ = 21) based on their GCS, with a significant difference noted in serum levels of MIF, MBP, and BDNF among patients with different degrees of severity (all $P$ < 0.001). The MIF, MBP, and BDNF levels were lower in the mild group compared to the moderate (all $P$ < 0.001) and severe group (all $P$ < 0.001). Additionally, the MIF and BDNF levels in the moderate group were lower compared to the severe group ($P$ = 0.011, $P$ = 0.002). Patients with mild severity had lower serum MIF, MBP, and BDNF levels than those with other degrees, and these indexes were positively correlated with the severity of TBI (all $P$ < 0.001, r = 0.62, r = 0.48, r = 0.58). Based on the GOS, patients were divided into a good prognosis group ($n$ = 107) and a poor prognosis group ($n$ = 24), with the levels of MIF, MBP, and BDNF in the good prognosis group being significantly lower than those in the poor prognosis group ($P$ < 0.001, $P$ = 0.007, $P$ = 0.003). The area under the curve (AUC) of MIF was higher than that of MBP and BDNF in predicting the prognosis of TBI patients; however, the statistical differences were not significant (MIF $vs$. MBP, $P$ = 0.239; MIF $vs$. BDNF, $P$ = 0.211; BDNF $vs$. MBP, $P$ = 0.899). The center line has a large displacement, CT annular cisterna compression, increased white blood cell count, MBP and BDNF were risk factors for prognosis in TBI patients ($P$ = 0.005, $P$ = 0.001, $P$ = 0.005, $P$ = 0.033, $P$ = 0.044).

**Conclusion:** The serum levels of MIF, MBP, and BDNF in TBI patients were positively correlated with the severity of the disease, and MBP, BDNF levels had predictive value in determining patient prognosis.

Corresponding author
Chengwu Liu,
15192005656@163.com

## INTRODUCTION

Traumatic brain injury (TBI) is an acquired and accidental brain injury that can result in various cognitive, sensory, executive, and mental impairments. The extent and location of brain injury largely dictate the impact of TBI on the body. Due to its rapid progression, several complications, and a high mortality rate, TBI poses significant threats to both an individual's life and health (*Saengrung et al., 2022*). The pathogenesis of craniocerebral injury is quite complex; TBI can cause damage to brain tissue, disrupt the blood-brain barrier, and trigger local brain edema through multiple mechanisms, leading to several secondary insults which worsen the disease condition. Although significant strides have been made in treating TBI, and there have been significant improvements in clinical outcomes, the mortality rate among critically-ill patients remains high (*Hägglund, Olivecrona & Koskinen, 2022*). Inflammation is observed to be significantly related to the prognosis of TBI patients (*Klein et al., 2022*). Macrophage migration inhibitory factor (MIF), a proinflammatory factor, plays a vital role in stimulating macrophage adhesion and phagocytosis, cytokine production, and inhibiting macrophage migration. It is mainly secreted by T cells, but monocytes/macrophages and adenohypophysis cells can also secrete MIF protein (*Zhou et al., 2019*). Glial cells can also secrete MIF in the CNS; overexpression of MIF is reported to be significantly increased in pathological conditions (*da Silva et al., 2019*). Myelin basic protein (MBP), is a specific protein synthesized by oligodendrocytes of the vertebrate CNS and Schwann cells in the peripheral nervous system. MBP exists in the CNS, including the brain, spinal cord, and cerebrospinal fluid. In other tissues, MBP exists at lower concentrations, making it difficult to measure (*Lashkarivand et al., 2020*). Research has suggested that large quantities of MBP in brain tissue can enter cerebrospinal fluid when the disease leads to the destruction of nerve tissue cells and myelin sheath, and when the blood-brain barrier is breached, MBP is released into the blood in large amounts, causing a significant increase in its concentration (*Lashkarivand et al., 2020*). Brain-derived neurotrophic factor (BDNF), an essential member of neurotrophic factors, can induce changes in neuronal structure and function by stimulating the hippocampal amygdala and prefrontal cortex. The hypothalamic-pituitary and epinephrine pathways are also affected, resulting in alterations in the structure and function of neurons (*Ye et al., 2019*). Based on this, this study aims to evaluate the association between serum levels of MIF and both the condition and prognosis of TBI patients. Our goal is to provide a reference for assessing disease condition and evaluating patient prognosis.

## MATERIALS AND METHODS

### Clinical materials

This study was a retrospective study and was approved by the Research Ethics Committee of the Qingdao Chengyang District People's Hospital. Inclusion criteria: Patients (1) with craniocerebral injury caused by accident and confirmed by head CT; (2) admitted within 12 h of injury; (3) aged ≥18 years; and (4) with complete clinical data available. Exclusion criteria: (1) patients with multiple limb fractures; (2) patients with immunological diseases; (3) patients with malignant tumors; (4) patients with pre-existing neurological disorders; (5) patients with space-occupying lesions in the craniocerebral region; (6) those with a history of alcohol or drug addiction; (7) those combined with severe kidney dysfunction; and (8) patients with previous hemorrhagic or ischemic stroke; (9) patients with hypertension or diabetes.

### Methods

#### CT diagnose

The GE16-row spiral CT scanner was used for axial scanning. The scanning baseline was the connection between the outer canthus and the inner auditory hilum, the tube current was 125 mA, the tube voltage was 125 kV, the layer spacing was 10 mm, the layer thickness was 10 mm, the matrix was $512 \times 512$, the scanning time was 3 s, the thin layer scanning was used, the layer thickness and the layer distance were both 5 mm. Images were processed by extended Brilliance Workspace V4.5.2 software to obtain the size of craniocerebral hematoma, estimate the amount of blood loss and read the midline location offset and lateral ventricle width to confirm craniocerebral injury.

#### Severity grouping

The Glasgow Coma Scale (GCS) was used to evaluate the severity of patients after admission (*Mikolić et al., 2021*), and they were divided into mild group (13–15 points), moderate group (9 ≤ 13 points), and severe group (3 ≤ 9 points).

#### Prognosis grouping

After 6 months of follow-up, patients were grouped according to the Glasgow Outcome Scale (GOS) (*Ingram et al., 2023*). The patients with GOS ≤3 were classified as having poor prognosis, and those with GOS scores of 4–5 were considered to have good prognosis.

#### BDNF, MIF and MBP and baseline data collection

Baseline data and laboratory indicators (serum BDNF, MIF, MBP, White blood cell count, APTT, PT and Fib) at admission of all patients were collected. The BDNF and MIF kits were provided by Wuhan BOSTER Biological Technology Co., LTD., (China), and the MBP kit was provided by Shanghai Sinovac Biomedical Research Institute.

### Observational indexes

(1) Severity of disease: The levels of serum BDNF, MIF and MBP in mild, moderate and severe groups were compared, and the correlation between each index and severity of disease was analyzed. (2) Prognosis: Serum BDNF, MIF and MBP levels were compared

between the poor prognosis group and the good prognosis group, and the predictive value of single serum index detection and combined detection on prognosis was analyzed.

## Statistical treatment

The data obtained were processed by SPSS 22.0 software. The counting data were expressed in % and compared by $\chi^2$ test. The measurement data were expressed by $(\bar{x} \pm s)$ after normality test. $T$ test was used to compare the differences between two groups, and univariate analysis test was used to compare the differences between multiple groups. Bonferroni method was used to correct the $P$ value. Spearman test was used to analyze the correlation between serum BDNF, MIF and MBP levels and the severity of TBI. Receiver operator characteristic (ROC) curve was used to analyze the predictive value of serum BDNF, MIF and MBP levels alone and in combination for the prognosis of TBI patients. Logistic regression was employed to analyze the relationship between serum BDNF, MIF and MBP levels and poor prognosis in patients with TBI patients. $P$ values were calculated by double side, $P < 0.05$ was considered statistically significant.

## RESULTS

The clinical data of 131 patients with traumatic brain injury from February 2019 to January 2022 were retrospectively analyzed, including 90 males and 41 females. The average age was $50.86 \pm 6.03$ years old. The causes of trauma included traffic accident in 61 cases, degradation in 45 cases, percussive injury in 16 cases, and others in nine cases.

### Comparison of clinical data of TBI patients with different severity

Patients were divided into mild group ($n = 63$), moderate group ($n = 47$) and severe group ($n = 21$) according to GCS score. There were no significant differences in gender, age, BMI, systolic blood pressure, diastolic blood pressure, heart rate trauma cause, CT annular pool status, APTT, PT and Fib among patients with different severity ($P = 0.560$, $P = 0.247$, $P = 0.105$, $P = 0.200$, $P = 0.974$, $P = 0.627$). $P = 0.999$, $P = 0.707$, $P = 0.100$, $P = 0.398$, $P = 0.103$, there were statistically significant differences in midline displacement and WBC count among the three groups (all $P < 0.001$; Table 1).

### Comparison of serum MIF, MBP and BDNF levels in patients with different severity

The levels of serum MIF, MBP and BDNF in patients with different severity were significantly different (all $P < 0.001$). The levels of MIF, MBP and BDNF in mild group were lower than those in moderate group (all $P < 0.001$) and severe group (all $P < 0.001$), and MIF and BDNF levels in moderate group were lower than those in severe group ($P = 0.011$, $P = 0.002$). The serum levels of MIF, MBP and BDNF in patients with mild degree were lower than those in patients with other degrees (Fig. 1).

### Correlation analysis of serum MIF, MBP and BDNF levels with disease severity

Serum levels of MIF, MBP and BDNF were positively correlated with the severity of TBI (all $P < 0.001$, r = 0.620, r = 0.480, r = 0.575; Fig. 2).

**Table 1 Comparison of clinical data of TBI patients with different severity.**

| Index | | Mild group ($n = 63$) | Moderate group ($n = 47$) | Severe group ($n = 21$) | $F/\chi^2$ | $P$ |
|---|---|---|---|---|---|---|
| Gender | Mela | 46 | 31 | 13 | 1.161 | >0.05 |
| | Female | 17 | 16 | 8 | | |
| Age (years) | | $51.02 \pm 5.97$ | $50.74 \pm 5.08$ | $50.62 \pm 4.39$ | 0.061 | >0.05 |
| BMI (kg/m$^2$) | | $22.85 \pm 2.41$ | $22.71 \pm 2.86$ | $22.93 \pm 2.16$ | 0.069 | >0.05 |
| Systolic blood pressure (mmHg) | | $134.19 \pm 9.45$ | $133.26 \pm 9.32$ | $132.58 \pm 10.45$ | 0.275 | >0.05 |
| Diastolic blood pressure (mmHg) | | $81.27 \pm 4.93$ | $81.74 \pm 4.58$ | $81.03 \pm 5.06$ | 0.206 | >0.05 |
| Heart rate (times/min) | | $85.44 \pm 4.73$ | $84.91 \pm 4.56$ | $85.24 \pm 4.12$ | 0.185 | >0.05 |
| Causes of trauma | Traffic accident | 29 | 22 | 10 | 0.518 | >0.05 |
| | Degenerate | 21 | 17 | 7 | | |
| | Percussive injury | 8 | 5 | 3 | | |
| | Others | 5 | 3 | 1 | | |

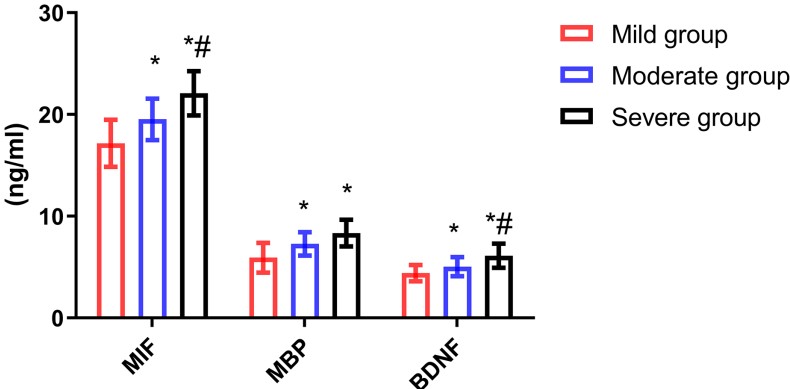

**Figure 1 Comparison of serum MIF, MBP and BDNF levels in patients with different severity.** Compared with mild group, $^*P < 0.05$; compared with moderate group, $^\#P < 0.05$.

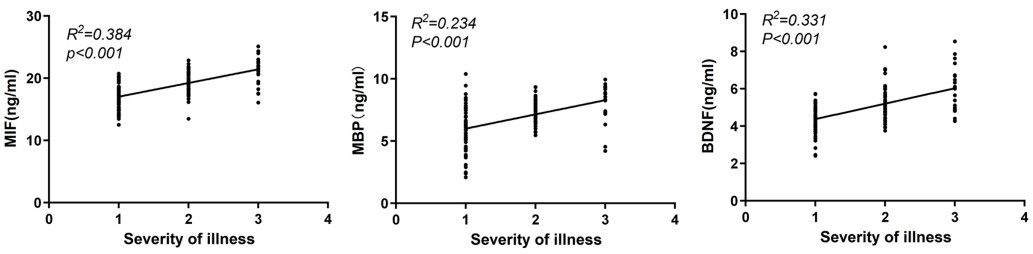

**Figure 2 Correlation analysis of serum MIF, MBP and BDNF levels with disease severity.**

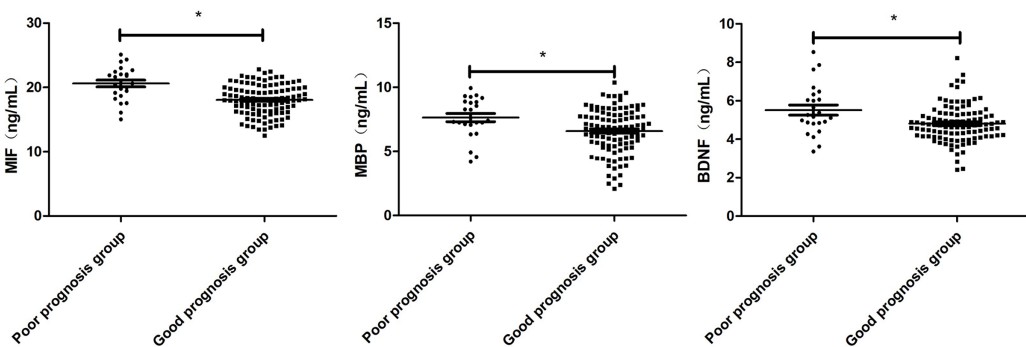

**Figure 3 Comparison of Serum MIF, MBP and BDNF Levels in patients with good prognosis and patients with poor prognosis.** Comparison between the two groups, *$P < 0.05$.

**Table 2 Prognostic value of serum MIF, MBP and BDNF levels.**

| Index | Cutoff value | AUC | SE | 95% CI |
|---|---|---|---|---|
| BDNF | 19.18 ng/mL | 0.672* | 0.064 | [0.545–0.798] |
| MBP | 7.04 ng/mL | 0.682* | 0.060 | [0.564–0.801] |
| MIF | 4.95 ng/mL | 0.767 | 0.056 | [0.656–0.877] |

**Note:**
Compared with MIF, *$P < 0.05$.

## Comparison of serum MIF, MBP and BDNF levels in patients with good prognosis and patients with poor prognosis

The patients were divided into good prognosis group ($n = 107$) and poor prognosis group ($n = 24$) according to GOS score. The levels of MIF, MBP and BDNF in the group with good prognosis were lower than those in the group with poor prognosis ($P < 0.001$, $P = 0.007$, $P = 0.003$; Fig. 3).

## Prognostic value of serum MIF, MBP and BDNF levels

The AUC of MIF in predicting the prognosis of TBI patients was greater than that of MBP and BDNF, but the difference was not statistically significant, MIF~MBP ($P = 0.239$), MIF~BDNF ($P = 0.211$), BDNF~MBP ($P = 0.899$) (Table 2 and Fig. 4).

## Comparison of clinical data of TBI patients in good prognosis group and poor prognosis group

There were statistically significant differences between the good prognosis group and the poor prognosis group in Center line displacement, CT ring pool status, White blood cell count, APTT, PT and Fib ($P < 0.001$, $P = 0.005$, $P < 0.001$, $P = 0.005$, $P = 0.002$; Table 3).

## Logistic regression analysis of prognosis in patients with TBI

The center line has a large displacement, CT annular cisterna compression, increased white blood cell count, MBP and BDNF were risk factors for prognosis in TBI patients ($P = 0.005$, $P = 0.001$, $P = 0.005$, $P = 0.033$, $P = 0.044$; Tables 4 and 5).

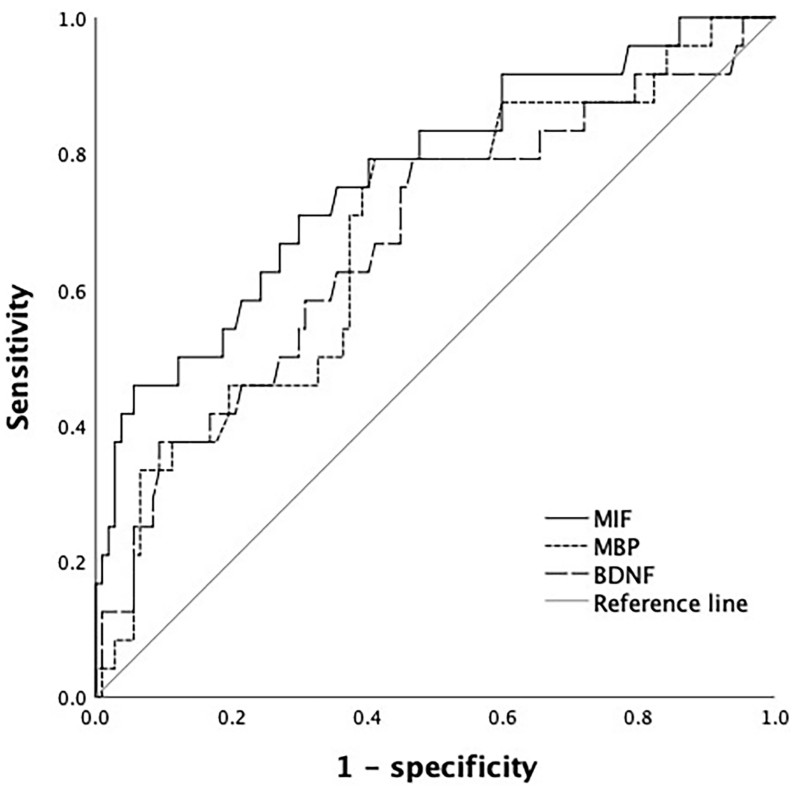

**Figure 4 ROC curve of serum MIF, MBP and BDNF levels in predicting prognosis of patients.**

**Table 3 Logistic regression analysis of serum MIF, MBP and BDNF levels and prognosis of patients.**

| Index | β | SE | Wald $\chi^2$ | OR | 95% CI | P |
|---|---|---|---|---|---|---|
| BDNF | 0.193 | 0.245 | 0.621 | 1.213 | [0.750–1.961] | 0.431 |
| MBP | 0.217 | 0.177 | 1.503 | 1.242 | [0.878–1.758] | 0.221 |
| MIF | 0.368 | 0.130 | 8.013 | 1.445 | [1.120–1.864] | 0.005 |
| Constant | −1.116 | 0.244 | 20.919 | 0.328 | [0.203–0.528] | <0.001 |

**Note:**
Assignment: Prognosis: (1 for poor prognosis, 0 for good prognosis); BDNF (≥19.18 ng/mL = 1, <19.18 ng/mL = 0); MBP (≥7.04 ng/mL = 1, <7.04 ng/mL = 0); MIF (≥4.95 ng/mL = 1, <4.95 ng/mL = 0).

# DISCUSSION

TBI, a complex injury with multiple causes, can result in significant pathological changes and functional abnormalities in the brain tissue (*Ritchie & Slomine, 2022*). However, the underlying mechanisms responsible for this still remain unclear. A growing body of research suggests that inflammatory state changes play a critical role in the progression of TBI, where inflammatory factors can exacerbate neuronal injury and nervous system deterioration (*Couch & Stewart, 2022*). The macrophage migration inhibitory factor (MIF) is a T-cell cytokine primarily secreted by the pituitary gland, as well as circulating macrophages and monocytes. Several recent studies have revealed that MIF acts as a proinflammatory cytokine implicated in the pathophysiology of other inflammatory

**Table 4 Assignment table.**

| Factor | Assign |
|---|---|
| Prognosis | Good prognosis = 0, poor prognosis = 1 |
| Center line displacement | ≤9 mm = 0, >9 mm = 1 |
| CT ring pool status | Normal = 0, pressure = 1 |
| White blood cell count | ≤15 × 109/L = 0, >15 × 109/L = 1 |
| APTT | ≤37 s = 0, >37 s = 1 |
| PT | ≤14 s = 0, >14 s = 1 |
| Fib | ≤2.91 g/l = 0, >2.91 g/l = 1 |
| MIF | ≤19.45 ng/mL = 0, >19.45 ng/mL = 1 |
| MBP | ≤7.06 ng/mL = 0, >7.06 ng/mL = 1 |
| BDNF | ≤4.78 ng/mL = 0, >4.78 ng/mL = 1 |

**Table 5 Logistic regression analysis of serum MIF, MBP and BDNF levels and prognosis of patients.**

| Index | β | SE | Wald χ2 | OR | 95% CI | P |
|---|---|---|---|---|---|---|
| Center line displacement | 2.630 | 0.944 | 7.575 | 13.872 | [2.180–88.282] | 0.005 |
| CT ring pool status | 3.870 | 1.186 | 10.643 | 47.956 | [4.688–490.524] | 0.001 |
| White blood cell count | 2.666 | 0.940 | 8.045 | 14.383 | [2.279–90.774] | 0.005 |
| APTT | 1.550 | 0.860 | 3.248 | 4.712 | [0.873–25.433] | 0.072 |
| PT | 1.040 | 0.876 | 1.408 | 2.828 | [0.508–15.745] | 0.235 |
| Fib | −0.260 | 0.845 | 2.550 | 0.771 | [0.147–4.038] | 0.758 |
| MIF | 1.548 | 0.969 | 2.550 | 4.700 | [0.703–31.403] | 0.110 |
| MBP | 2.195 | 1.027 | 4.565 | 8.983 | [1.199–67.296] | 0.033 |
| BDNF | 2.327 | 1.158 | 4.038 | 10.247 | [1.059–99.169] | 0.044 |
| Constant | −11.336 | 2.505 | 20.478 | 0.000 | | <0.001 |

diseases, such as sepsis and autoimmune liver disease. Brain-derived neurotrophic factor (BDNF), a crucial neurotrophin family member, can stimulate the amygdala and prefrontal cortex of hippocampus through various mechanisms, leading to structural and functional changes in neuroplasticity, which affect the hypothalamic-pituitary and epinephrine pathways (*Fan et al., 2022*). Some studies show that BDNF is more neurotrophic in the brain, providing significant protection to both central and peripheral nerve cells, promoting effective repair in case of damage and normal functioning (*Lindholm & Saarma, 2022*). MBP, a specific protein of oligodendrocytes and other neural tissues found in the vertebrate central nervous system, is released in large quantities into cerebrospinal fluid and then, into the blood after nerve tissue cells and myelin sheath are damaged or involved due to traumatic brain injury, often leading to an increase in the serum MBP level (*Forsyth et al., 2022*). Studies reveal that serum MBP concentration is linked to the severity of the myelin injury of the white matter oligodendrocyte and destruction of the blood-brain barrier (*DiTommaso et al., 2022*). The current study analyzes the relationship between MBP, MIF and BDNF, and TBI severity, and reveals that

the levels of MIF, MBP and BDNF in the mild group were lower than those in the moderate and severe groups. Notably, the levels of these biomarkers in the moderate group were lower than those in the severe group, suggesting their correlations with TBI severity. Further investigation confirms that serum levels of MIF, MBP and BDNF are positively related to TBI severity. The more severe the patient's condition, the higher the levels of these biomarkers. Interestingly, BDNF can protect central and peripheral nerves and repair injured nerve cells. However, the blood-brain barrier's damage results in decreasing BDNF levels in brain tissue, which hinders the recovery of the patient's neurological function. Overexpression of MIF can act as a chemokine-like cytokine, recruiting leukocytes to the inflammatory site, directly promoting the production or release of a large number of proinflammatory molecules. Moreover, exogenous MIF supplementation can cause nerve cell death in TBI rats, leading to neurological impairment, which might exacerbate the degree of neurological dysfunction. Overall, serum MBP concentration reflects the degree of blood-brain barrier damage and may thus serve as an indicator of the degree of craniocerebral injury (*Abdulla et al., 2022*).

TBI patients may experience blood-brain barrier disorders, which could result in secondary injury leading to exposure of brain parenchyma and other pathological changes, including brain edema, neurological dysfunction, and inflammation. These changes may significantly impact the prognosis of patients (*Čekanauskaitė et al., 2020*). The statistics suggest that most TBI patients are likely to suffer central nervous system injury, and in extreme cases, they may die (*Jaudon et al., 2021*). BDNF could reach damaged neurons through autocrine or paracrine pathways to bind specifically to TrkB receptors, regulating synaptic transmission and promoting neuronal repair. When these neurons undergo damage, BDNF gets released into the cerebrospinal fluid and enters the blood circulation *via* the blood-brain barrier, dramatically increasing peripheral blood levels. MBP is a crucial protein present in myelin, which is primarily damaged in the central nervous system. Such damage leads to alterations in the function of the blood-brain barrier, increasing serum MBP level in the patients. The inflammatory reaction is fundamental to the pathophysiology of the TBI as the central nervous system's superposition and mutual deterioration of the local and systemic inflammatory response can worsen cerebral nerve function injury and lead to the compromised prognosis of TBI patients (*Sarkis et al., 2022*). As an immunocompetent and typical cytokine of inflammation, Macrophage MIF fosters the continuous development of diseases *via* cell proliferation and angiogenesis stimulation. MIF functions as a chemokine-like cytokine that recruits white blood cells to the inflammatory site, promoting the production or release of various inflammatory molecules directly or indirectly. The study shows that the levels of MIF, MBP, and BDNF were relatively lower in the better prognosis group than in the poor outcome group. Thus, each index can have correlations with the prognosis of patients.

Currently, there is a lack of objective biochemical indicators to evaluate the conditions and prognosis of patients with craniocerebral injury early on. Relevant studies suggest that accurately gauging the patient's condition and conducting preliminary prognosis evaluations is crucial to developing relevant treatment plans in the early stages of the disease (*Kowalski, Whyte & Giacino, 2021*). The present study indicates that the serum

levels of MIF, MBP and BDNF correlate with the disease's severity, and the serum level of each indicator differs in patients with contrasting prognoses. The study posits that the serum levels of MIF, MBP, and BDNF could be utilized to evaluate patient prognosis. Reports suggest that MBP and BDNF in cerebrospinal fluid and blood are specific biochemical markers of acute brain damage and acute demyelination, respectively (He et al., 2019). Brain injury leads to severe damage, affecting not only the brain's gray matter and neurons but also demyelinating the white matter, increasing MBP and BDNF concentrations in cerebrospinal fluid. Brain injury also damages the blood-brain barrier or alters its permeability, resulting in a significant increase in serum MBP and BDNF concentrations. Furthermore, the study's results revealed that MBP and BDNF were risk factors for prognosis in TBI patient, this may be because the higher the content of MBP and BDNF, the more serious the nervous system damage and the worse the prognosis of patients. It's physiological mechanism needs to be further explored.

Previous animal experiments have reported that the gene knockout of MIF could reduce nerve cell death and promote nerve function recovery in TBI mice. In contrast, MIF supplementation caused nerve cell die-off leading to neurological impairment in TBI rats (Lee et al., 2020). However, few studies have examined serum MIF level changes in TBI patients. In the present study, the serum MIF level in TBI patients was measured, and its relationship with severity was further evaluated. The research found that the serum MIF level of TBI patients correlated with the disease's severity, revealing its potential as an excellent biomarker for assessing TBI patients' conditions. This study observed that MIF's area under the curve (AUC) value for predicting TBI prognosis was greater than that of MBP and BDNF, exceeding 0.75 in value, indicating that MIF had prognostic value for patients. Therefore, serum MIF levels could serve as an auxiliary evaluation method for patients.

## CONCLUSIONS

The serum concentrations of MIF, MBP, and BDNF in TBI patients exhibited a positive correlation relative to the disease's severity, and MBP, BDNF levels had reliable prognostic capabilities for patient outcomes. While this study demonstrates prognostic potential, there are shortcomings in the approach. The study's retrospective design and low sample size in the analyzed data present the possibility of biased results. Consequently, further prospective analyses with larger sample sizes may be necessary to understand this relationship thoroughly.

### Funding

The authors received no funding for this work.

### Competing Interests

The authors declare that they have no competing interests.

## Author Contributions

- Zhentong Liu conceived and designed the experiments, performed the experiments, prepared figures and/or tables, authored or reviewed drafts of the article, and approved the final draft.
- Chengwu Liu conceived and designed the experiments, analyzed the data, prepared figures and/or tables, and approved the final draft.
- Kegao Ma conceived and designed the experiments, performed the experiments, analyzed the data, authored or reviewed drafts of the article, and approved the final draft.

## Human Ethics

The following information was supplied relating to ethical approvals (*i.e.*, approving body and any reference numbers):

This study was approved by the Ethics Committee of Qingdao Chengyang District People's Hospital.

## Data Availability

The raw data is available in the Supplemental Files.

## Supplemental Information

Supplemental information for this article can be found online at http://dx.doi.org/10.7717/peerj.15933#supplemental-information.

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
