# Peer review of "Retrospective study on the correlation between serum MIF level and the condition and prognosis of patients with traumatic head injury"

_PeerJ, doi:10.7717/peerj.15933_

## Round 0.1 · original submission · Major Revisions

The correlation of serum MIF, MBP and BDNF levels with disease and prognosis in patients with traumatic brain injury is fascinating. But the authors' statistical results need to be refined and the generalisability of their conclusions explained. In addition, the inclusion criteria of patients were further clarified.

·

Basic reporting

1)Did patients with diabetes were included?
2)Each p value should be given in the manuscript.
3)“The levels of serum MIF, MBP and BDNF in patients with different severity were significantly different (all P<0.05). The levels of MIF, MBP and BDNF in mild group were lower than those in moderate group and severe group, and the levels in moderate group were lower than those in severe group (all P<0.05; Figure 1).” This presentation was too abstract. Please state which extent did it lower than others.

Experimental design

4)“with craniocerebral injury caused by accident and confirmed by head CT” Which kind of CT and which parameters were used.
5)“with complete clinical data” How to understand the “complete clinical data”? Did this study a retrospective study?
6)Please state potential limitations in this study.
7)“This study is a retrospective study with a small number of cases included in the analysis, which may lead to bias in the results.” Why did a retrospective study could collect the serum from involved patients?
8)“with craniocerebral injury caused by accident and confirmed by head CT” How to confirm the craniocerebral injury?

Validity of the findings

no comment

Additional comments

no comment

Reviewer 2 ·

Basic reporting

The basic reporting in the paper is reasonable with enough literature references and background/context provided.

Experimental design

The primary concern in this paper is the experiment design and analysis. The paper investigated the correlations between Serum BDNF, MIF, and MBP levels and Glasgow prognostic score (GOS). The paper arbitrarily grouped serum BDNF, MIF, and MBP levels and the GOS to determine the significant correlations. The problem with this is that papers failed to explain the impact of grouping arbitrarily on results, i.e. the results remain the same with slight changes in the grouping.

The significance values reported in the paper are barely crossing alpha values and are uncorrected for multiple comparisons, therefore, the validity of the results is questionable and they do not support the claims made.

Validity of the findings

Same as above.

·

Basic reporting

A. “The clinical data of 131 patients with traumatic brain injury from February 2019 to January 2022 were retrospectively analyzed, including 90 males and 41 females. The average age was 50.86±6.03 years old. The causes of trauma included traffic accident in 61 cases, degradation in 45 cases, percussive injury in 16 cases, and others in 9 cases.” This part should belong to “2 Results”.
B. Did patients with hypertension were included?
C. “with craniocerebral injury caused by accident and confirmed by head CT” How to confirm by head CT.
D. P values were calculated by single tail (side) or double tail (side) should be given.
E. “Serum levels of MIF, MBP and BDNF were positively correlated with the severity of TBI (all P<0.05; Figure 2).” Please further introduce this part with more details.
F. Why did MIF, MBP and BDNF were selected for this study?
G. “The measurement data were expressed by ( ± s) after normality test.” How to make normality test?

Experimental design

It is overall well and organized.

Validity of the findings

Well

Additional comments

None

---

## Round 0.2 · Minor Revisions

The language of the manuscript needs to be touched up. the inclusion of Logis regression analysis variables needs to be reconsidered. In addition, the innovative nature of the article's reviewer concerns needs to be explained by the authors.

·

Basic reporting

The article meets the PeerJ criteria.

Experimental design

The article meets the PeerJ criteria.

Validity of the findings

The article meets the PeerJ criteria.

Additional comments

The article meets the PeerJ criteria.

Reviewer 2 ·

Basic reporting

NA

Experimental design

The authors have indeed provided updated results and error corrections in the revised version. However, the absence of p-values with corrections for multiple comparisons or FDR still persists in the paper. Given the experiment's design flaws and insufficient and incorrect statistical analysis, there are valid concerns about the authenticity of the results. Therefore, it is my strong recommendation to reject this manuscript based on these significant shortcomings.

Validity of the findings

Same as above.

·

Basic reporting

Accept

Experimental design

Accept

Validity of the findings

Accept

Additional comments

Accept

Reviewer 4 ·

Basic reporting

This is a valuable article. I only mention the most important major ones here:
1 The title should indicate the aim and conclusion in a brief way.
2 For prognosis of patients with traumatic brain injury, is there any innovation in this article compared to the previous article?
3. The results of MIF, MBP and BDNF should be further discussed to support the conclusion.
4. The severity of disease is not clear from the narrative page 3; many sentences are missing.
5. Logistic regression analysis of serum MIF, MBP and BDNF levels and prognosis of patients not detailed. Correction for covariates, which ones?
6. Generally, the moderate group should be named middle group instead of moderate group, pls modify it in the MS.
7. The introduction of n why this article chooses MIF, MBP and BDNF for research is too general. Please describe the specific problems to be solved for traumatic brain injury.

Experimental design

This is a valuable article. I only mention the most important major ones here:
1 The title should indicate the aim and conclusion in a brief way.
2 For prognosis of patients with traumatic brain injury, is there any innovation in this article compared to the previous article?
3. The results of MIF, MBP and BDNF should be further discussed to support the conclusion.
4. The severity of disease is not clear from the narrative page 3; many sentences are missing.
5. Logistic regression analysis of serum MIF, MBP and BDNF levels and prognosis of patients not detailed. Correction for covariates, which ones?
6. Generally, the moderate group should be named middle group instead of moderate group, pls modify it in the MS.
7. The introduction of n why this article chooses MIF, MBP and BDNF for research is too general. Please describe the specific problems to be solved for traumatic brain injury.

Validity of the findings

This is a valuable article. I only mention the most important major ones here:
1 The title should indicate the aim and conclusion in a brief way.
2 For prognosis of patients with traumatic brain injury, is there any innovation in this article compared to the previous article?
3. The results of MIF, MBP and BDNF should be further discussed to support the conclusion.
4. The severity of disease is not clear from the narrative page 3; many sentences are missing.
5. Logistic regression analysis of serum MIF, MBP and BDNF levels and prognosis of patients not detailed. Correction for covariates, which ones?
6. Generally, the moderate group should be named middle group instead of moderate group, pls modify it in the MS.
7. The introduction of n why this article chooses MIF, MBP and BDNF for research is too general. Please describe the specific problems to be solved for traumatic brain injury.

Additional comments

This is a valuable article. I only mention the most important major ones here:
1 The title should indicate the aim and conclusion in a brief way.
2 For prognosis of patients with traumatic brain injury, is there any innovation in this article compared to the previous article?
3. The results of MIF, MBP and BDNF should be further discussed to support the conclusion.
4. The severity of disease is not clear from the narrative page 3; many sentences are missing.
5. Logistic regression analysis of serum MIF, MBP and BDNF levels and prognosis of patients not detailed. Correction for covariates, which ones?
6. Generally, the moderate group should be named middle group instead of moderate group, pls modify it in the MS.
7. The introduction of n why this article chooses MIF, MBP and BDNF for research is too general. Please describe the specific problems to be solved for traumatic brain injury.

Reviewer 5 ·

Basic reporting

The manuscript should be edited by a professional English speaker to improve the logic and minimize grammar mistakes.

Experimental design

In the logistic analysis, only BDNF, MIF, and MBP levels were included. I believe other variables, such as age, gender, and other demographic features should also be included.

Table one exhibited clinical data of TBI patients in this study. How about clinical data concerning treatment, and imaging features, and other blood test indicators (such as white cell acount, coagulation indexes) that may influence the prognosis of the TBI patients?

Validity of the findings

Conclusions are well stated, linked to original research question & limited to supporting results.

Additional comments

None

---

## Round 0.3 · accepted · Accept

The author addressed the reviewer's concerns.

·

Basic reporting

no comment

Experimental design

no comment

Validity of the findings

no comment

Additional comments

no comment

·

Basic reporting

No more comments.

Experimental design

No more comments.

Validity of the findings

No more comments.

Additional comments

No more comments.

Reviewer 4 ·

Basic reporting

authors 's revision is good

Experimental design

authors 's revision is good

Validity of the findings

authors 's revision is good

Additional comments

authors 's revision is good

Reviewer 5 ·

Basic reporting

The authors critically answered the questions I raised before. I think this manuscript could be accepted at this version.

Experimental design

no comment

Validity of the findings

no comment

Additional comments

no comment